# Healthy or Not Healthy? A Mixed-Methods Approach to Evaluate Front-of-Pack Nutrition Labels as a Tool to Guide Consumers

**DOI:** 10.3390/nu14142801

**Published:** 2022-07-08

**Authors:** Melvi Todd, Timothy Guetterman, Jako Volschenk, Martin Kidd, Elizabeth Joubert

**Affiliations:** 1Department of Food Science, Stellenbosch University, Stellenbosch 7599, South Africa; joubertL@arc.agric.za; 2Department of Family Medicine, University of Michigan, Ann Arbor, MI 48104, USA; tguetter@med.umich.edu; 3Department of Economic and Management Sciences, Stellenbosch Business School, Stellenbosch University, Bellville 7530, South Africa; jakov@usb.ac.za; 4Centre for Statistical Consultation, Department of Statistics and Actuarial Sciences, Stellenbosch University, Stellenbosch 7599, South Africa; mkidd@sun.ac.za; 5Plant Bioactives Group, Post-Harvest and Agro-Processing Technologies, Agricultural Research Council, Infruitec-Nietvoorbij, Stellenbosch 7599, South Africa

**Keywords:** FOP nutrition label, consumer, health promotion, nutritional policy, non-communicable disease, ultra-processed food, mixed methods

## Abstract

This study explored how South African food labels could be improved, to enhance customer evaluation of the overall healthiness of packaged food. Focus was given to the comparison of front-of-pack (FOP) nutrition labels as a quick assessment tool. The exploratory sequential mixed-methods design used qualitative interviews (*n* = 49) to gain insight into labeling challenges and select FOP nutrition labels for consumer testing. Consumers (*n* = 1261) randomly assessed two out of six possible FOP nutrition labels relative to a ‘no-label’ control in one of 12 online surveys, applied to a fictitious cereal product. A mixed-model analysis of variance was used to compare the differences in health ratings for the different FOP nutrition labels. The interviews revealed three themes for label improvement, that are presented over three time horizons. In terms of helping consumers identify less healthy products, the effect sizes were most prominent for health warnings (*p* < 0.01) and low health star ratings (*p* < 0.01). The findings of this research not only clarify whether FOP nutrition labeling formats used in other regions such as Europe, South America and Australia could be useful in the South African context, but they can assist policymakers and decision-makers in selecting an effective FOP label.

## 1. Introduction

In 2016, non-communicable diseases (NCDs), such as diabetes, cancer, cardiovascular diseases, and chronic respiratory diseases, accounted for over 51% of deaths in South Africa [1], with poor health disproportionately affecting the socioeconomically disadvantaged [2,3]. These people, who comprise 84% of the country’s population, rely on public health care [3]. The development of NCDs is strongly linked to an unhealthy diet [4], with ultra-processed foods being a particular pain point [5]. Ultra-processed foods are typically high in ‘nutrients of concern’ (i.e., saturated fats, sugar and sodium), which have been implicated in the development of NCDs [5,6]. Ultra-processed foods are typically energy dense with low nutritional value, but they are frequently substantially cheaper than healthier alternatives [7], making it no coincidence that the burden of diet-related NCDs is shifting towards the socioeconomically disadvantaged [8]. Furthermore, large variations in serving size can make it tricky for consumers to accurately identify healthy and less healthy foods [9]. This technique of reporting nutritional information according to varying serving sizes intentionally blurs the nutritional values of food to stimulate sales [9]. Furthermore, such distortions of information also make it more difficult for consumers to control or limit their intake of ‘nutrients of concern’. Assisting consumers in identifying healthier foods with better nutritional profiles in this confusing landscape is thus a practical intervention point for preventing and curbing NCD progression [10].

Food labels are tools available to consumers to help them judge the nutritional quality of packaged food [11], potentially supporting more informed and healthier choices. Over the longer term, more frequent, healthier food choices can reasonably be expected to support a reduction in the risk of becoming obese or developing NCDs [10,12]. This link between food labels which inform consumers, the potential impact this has on their food choice and thus the longer term health implications (i.e., the development of obesity and NCDs) are the reason why food labels receive much attention in literature today [12].

Several studies have explored the barriers South African consumers experience when using food labels [10,13,14,15,16,17]. These studies found that the most common barriers to understanding and applying food labels are readability and comprehensibility, some of which may stem from the fact that the country has 11 official languages [18] and disparate levels of education [19]. These challenges around the understanding and use of food labels, combined with the country’s growing obesity and NCD burden, points to the obvious need to ‘decipher’ food labels for South African consumers and guide them towards the identification of healthier choices.

Voluntary front-of-pack (FOP) nutrition labels are increasingly being used to aid consumers in rapidly assessing the nutritional qualities of food [20,21]. FOP nutrition labels are categorized into several types, i.e., endorsement logos (e.g., Keyhole and Choices) [22,23]; summary indicator systems (Health Star Rating and Nutri-Score) [24,25]; and nutrient-specific warning labels (e.g., Chilean warning label and nutrient-specific interpretive labels) [26]. FOP nutrition labels are often voluntary (e.g., United Kingdom and France), but some countries have opted for mandatory FOP nutrition labels (e.g., warning labels in countries such as Chile and Brazil). While definitive evidence is lacking that FOP nutrition labels result in the purchase of healthier food (since the majority of studies are experimental) [20,21,27], they do assist consumers in identifying healthier foods [28,29]. Furthermore, long-term evidence, such as that from the Dutch Choices program, illustrates significant changes in the composition of ultra-processed food (as a result of reformulation) over a ten-year period [30]. These changes include a reduction in salt, sugar, and saturated fat content, as well as an increase in fiber content.

South Africa has no mandatory FOP nutrition label. However, when browsing products on supermarket shelves, it becomes apparent that many multinational companies are including Guideline Daily Amounts (GDAs) on pack. In addition, imported foods carrying a Nutri-Score label are appearing with increasing frequency and many local manufacturers and supermarket chains have developed their own FOP nutrition labels. Considering global trends to recommend or mandate the use of FOP labels in the fight against obesity and NCDs [10,12,20,21] and heightened consumer exposure to multiple FOP nutrition label formats, this study aimed to determine what labeling interventions help South African consumers to better comprehend the healthiness of packaged foods. This included investigating to what extent various FOP labels can help South African consumers judge the overall healthiness of a fictitious cereal product. A fictitious maize-based cereal was chosen, given that it is a staple carbohydrate in South Africa and prepackaged cereals represent a growing food category [31]. Packaged cereals can serve as a refuge for hidden sugars [32] and are considered as ultra-processed foods according to the NOVA classification [33].

An exploratory sequential mixed-methods design was selected to take advantage of the benefits of the integration of qualitative and quantitative data [34]. Integration allows the researcher to gain new and in-depth insights compared to what would be possible with quantitative or qualitative methods alone [35,36,37,38]. For example, Reyes et. al. [39] used a mixed-methods approach for the development of the Chilean FOP warning label. Interviewing a broad range of stakeholders yielded suggestions for improvement of food labels, and the resulting insights guided the selection and testing of the FOP nutrition labels.

The findings of this research not only clarify whether FOP labeling formats used in other regions such as Europe, South America and Australia could be useful in the South African context, but they can assist policymakers and decision-makers in selecting an effective FOP label. Furthermore, the study provides valuable insights on future research opportunities, including those which incorporate the use of technology. Lastly, the research also emphasizes the need for multi-stakeholder collaboration to craft a comprehensive policy strategy to fight NCDs in South Africa.

## 2. Materials and Methods

### 2.1. Study Design and Setting

A pragmatic paradigm was adopted for this study, as is customary for mixed-methods research [35,40,41]. The steps followed in the exploratory sequential mixed-methods design are illustrated in Figure 1, which broadly incorporated a qualitative stage followed by a quantitative stage. In-depth interviews (qualitative stage) were conducted with a focused sample of South African professional and consumer participants to gain diverse inputs. The results from this qualitative stage of the study were used to construct a questionnaire (quantitative stage) that required consumers to compare and rate combinations of FOP labels on grounds of perceived healthiness. Both the in-depth interviews and the survey were conducted online due to COVID-19 pandemic restrictions during the data-collection period (April 2020 to May 2021).

### 2.2. Ethical Considerations

Ethical clearance for the project was obtained from Stellenbosch University before the commencement of the research. The research company enlisted for administration of the consumer survey strictly adhered to the Code of Conduct and Guidelines of the South African Marketing Research Association. This Code aims to ensure and maintain quality and professional research practice in Southern Africa. All respondent data were anonymized by the research company in accordance with the requirements for data handling stipulated in the Protection of Personal Information Act of South Africa.

### 2.3. Recruitment

#### 2.3.1. Interview Participants: Qualitative Study

Informed consent was obtained from all the participants prior to the start of each interview. Participants were thanked for their contributions but were not remunerated. A total of 49 participants were interviewed: 12 (24.5%) consumers and 37 (75.5%) professionals. The professional participants were specifically recruited on grounds of their diverse professional qualifications. Sampling was purposeful and targeted to achieve maximum variation in perspectives. The intention of the sampling was to include roughly three equal groups of professionals with tertiary qualifications from, respectively, (a) food manufacturing or retail, (b) healthcare, and (c) neither food nor healthcare. These categories were chosen based on possible conflicts of interest, and so that any food or healthcare biases could be detected. Ultimately, food-related professionals accounted for 35% (*n* =13) of the professional sample, followed by 32.5% (*n* = 12) healthcare-related professionals and 32.5% (*n* = 12) of those that were neither associated with food nor healthcare (Appendix A). Initially, professional contacts of the lead author were approached via LinkedIn. Subsequently, snowball sampling (i.e., chain-referral sampling) prevailed, accounting for approximately 50% of professional participants. Snowball sampling was used to produce a diverse sample of professional respondents that would be difficult to reach using other sampling methods [42].

#### 2.3.2. Survey Respondents: Quantitative Study

Consumer respondents were enlisted for the online survey from the database of a South African marketing research company. While respondents in the database had already consented to receive survey invitations via e-mail, their participation was still voluntary. Consumer respondents willing to participate in the survey followed the link to provide informed consent and answer the survey questions. Respondents were encouraged to participate by offering them an opportunity to win a share of ZAR 10,000 in prize money (±USD 650). The survey was open for two weeks in April 2021. Screening questions were included at the start of the survey to ensure that respondents (i) were responsible for the grocery purchases of a household (i.e., could exercise decision-making power on food purchases), (ii) purchased cereals, and (iii) were South African citizens. As a result of the screening, 664 responses were eliminated. A response rate of 10.8% was achieved. This equaled 1925 respondents of the total database of 17,822. Respondents who did not pass the screening questions were thanked and exited automatically from the survey; however, they were not excluded from the opportunity to win prize money. The marketing research company administrated the awarding of the prize money.

### 2.4. Procedure

#### 2.4.1. Interviews: Qualitative Study

Before the interviews, the interview guide was tested with a convenience sample of three professionals in the field of research. All interviews were conducted in English, and no translators were used. The interviews were conducted via video conferencing between February and May 2020. The duration of the interviews ranged from 45 to 60 min. Participants were requested to share their views on the challenges faced by South African consumers when using food labeling and potential ways in which labeling could be improved. Participants were allowed to answer the question freely, and based on their responses, follow-up questions were asked where additional insight or clarity was sought. For example, one of the interview questions was: ‘Do you believe that South African food labels are fit for purpose?’ Depending on the response, this could typically be followed up by “Why would you say that?” or “Can you provide an example?”. Interviews were completed until theoretical saturation was reached [43]; that is, when no additional themes emerged from the last five interviews, a final interview was conducted to confirm that saturation had been achieved. A similar process to confirm theoretical saturation has been followed in previous mixed-methods studies [44,45].

#### 2.4.2. Surveys: Quantitative Study

As previously mentioned, insights from the qualitative stage of the study were used to construct a quantitative questionnaire that required consumers to compare and rate the combinations of FOP labels. Owing to the large number of FOP labels to be evaluated (six), and to avoid respondent fatigue, 12 separate surveys were constructed, each comprising the same control (i.e., product without FOP label) and two of the six FOP labels (presented randomly). The details of the cereal packaging design and the six FOP labels are shown in Figure 2, and those of the FOP label configuration of the 12 surveys are shown in Figure 3.

The 12 permutations of the quantitative survey were piloted with ten individuals, including three staff members from the market research company. Minor adjustments were made to wording and alignment to ensure that images were displayed correctly on desktop computers and mobile devices. A tick box was added to each image, which needed to be checked before consumers could respond to questions. This failsafe was employed to ensure that the respondent ‘interacted’ with the survey before answering the corresponding questions.

The marketing research agency was responsible for balancing the administration of the 12 surveys with the database to ensure that the respondents were demographically similar. The surveys were open for two weeks in April 2021 and attracted 1261 valid responses.

In addition to rating the healthiness of the fictitious cereal product (1 = very unhealthy; 10 = very healthy), respondents were also asked to answer nine questions of the Health Consciousness Scale (HCS) [46]. The nine scores of the HCS were combined into an overall health consciousness score. From these data, it was possible to determine if a correlation existed between the ratings assigned to the fictitious cereal products (with differing FOP labels) and respondents’ level of health consciousness. Other health-related information sought from the respondents included their physical activity levels [47] and smoking status. The body mass indices (BMIs) of the respondents were calculated based on the height and weight data supplied, according to the formula:BMI = weight (kg)/[height (m)]^2^

### 2.5. Data Analysis

#### 2.5.1. Interview Data: Qualitative Study

The interviews were conducted by the lead author and audio-recorded with the participants’ permission, transcribed using Otter.ai (Otter.ai, Los Altos, CA, United States), with accuracy verified after transcription, and analyzed and coded using MAXQDA 2020 software (VERBI Software, Berlin, Germany). We followed the data-driven and inductive process of thematic analysis [48]. This process included accurate transcription with comprehensive coding that led to the generation of overarching themes and sub-themes. Open, in vivo coding allowed the participants’ views to give meaning to the data. Codes were arranged into groups with similar themes and named accordingly. Finally, the grouping of the latter sub-themes resulted in three, final overarching themes pertaining to food label improvement. A list of key challenges was compiled from interview feedback. 

#### 2.5.2. Survey Data: Quantitative Study

Descriptive statistics were used to define basic data features. Respondents were included as random effects, and labels and demographic variables were included as fixed effects. Normality was assessed by inspecting normal probability plots and judged to be acceptable. Fisher’s least significant difference (LSD) test was used for post-hoc testing. Cohen’s d effect sizes were calculated to assess the findings further. A mixed-model analysis of variance (ANOVA) was used to compare differences between the FOP label ratings of the fictitious packaged cereal product using Statistica (TIBCO Software, Palo Alto, CA, USA). Differences with a significance level of 5% (*p* < 0.05) were considered statistically significant.

#### 2.5.3. Integration of Data Sets

In this present study, a joint display was used to visually integrate [49] the quantitative and qualitative findings and to draw out new insights. 

### 2.6. Trustworthiness, Validity and Reliability

In the present study, trustworthiness was ensured through respondent validation [50,51]. Participants were requested to provide the authors with feedback on high-level findings (themes) and confirm that they were a true reflection of the interviews conducted. No changes were made to these themes. The research process was logical, traceable, and clearly documented, offering reliability. Finally, throughout the study, we motivated all methodological and analytical choices [52].

The experimental study (survey) was considered to have high internal validity because it could control for the investigated factors (i.e., different FOP labels). The reliability of the survey items was confirmed by determining the Cronbach’s alpha. In accordance with practice, a Cronbach alpha of 0.8 or greater was considered an indication of reliability [53,54].

## 3. Results

### 3.1. Study Sample Characteristics

#### 3.1.1. Qualitative Study (Interview Participants)

A total of 49 interviews were conducted with 37 professionals and 12 consumers. The details of the participants, including potential conflicts of interest between professionals working in healthcare or the food industry, are available in Appendix A.

#### 3.1.2. Quantitative Study (Survey Respondents)

Table 1 provides a characterization of the sample for the main sociodemographic variables, self-reported health circumstances, and food-shopping behavior. In total, 72.6% of respondents were women. The mean age of the respondents was 37 years (SD = 11.5; min = 18; max = 85), with a mean living situation being in a household with four individuals (SD = 2; min = 1; max = 10), one of which was a child (SD = 1, min = 0, max = 8). The average body mass index (BMI) of the sample was 29.4 (SD = 7.6; min = 14.5; max = 63). Despite a sample with an average BMI indicating an ‘overweight’ status, respondents had a high mean HCS score (35; SD = 8), indicating high health consciousness.

### 3.2. Qualitative Themes from the Interviews

Findings from the qualitative interviews were captured under three broad themes, namely ‘make it clearer’; ‘make it simpler’; and ‘make it smarter’, with subthemes reflecting the specific areas of improvement (illustrative quotes available in Appendix A). In addition to these improvement areas, key challenges related to physical and printed food labels were identified (Appendix A).

#### 3.2.1. Make it Clearer (Theme 1)

Three sub-themes related to making labels clearer to read and understand (Theme 1) were identified. These barriers hinder consumers’ ability to use labels, even if they are motivated to do so. The sub-themes included font size, legibility, and the use of plain language.

Font size and legibility are recurring problems in food labeling. Participants aptly noted that the average consumer needs to be equipped with a magnifying glass to read the information on pack, because the font size is ‘too small’. Consumers probably do not use the information on the pack because they cannot see it properly (too small and illegible). Increasing font size in proportion to pack size was suggested, as well as standardizing the color of the nutritional and ingredient information on the labels to black and white (for ingredients and nutritional information).

Another challenge highlighted by the participants is that the legibility criteria for food labeling in South Africa are fairly open to interpretation. Without clear criteria, no recourse is possible—whether or not the infringement is purposeful. Crisps and chocolates were frequently mentioned as offending product categories, with the legibility of relevant product information against a backdrop of metallized wrappers being problematic.

Finally, respondents indicated that food labels could be difficult to understand because of the use of scientific names for some ingredients. While food manufacturers may provide accurate information, more attention should be paid to the use of plain language, making it easier for consumers to interpret the label. A pertinent example includes the use of chemical names such as ‘sodium chloride’ instead of more colloquial terms, i.e., ‘salt’ or ‘table salt’.

#### 3.2.2. Make it Simpler (Theme 2)

Participants generally agreed that food labeling needed to be made simpler to use and understand. Recommendations included availability on the front of the packs, use of pictures, and limiting the use of color (i.e., simply using black and white). In terms of the more detailed ‘solutions’ suggested, there was a call to convert the key nutrient information into percentages, making it easier for consumers to understand what portion of their total daily requirement they are ingesting when consuming a specific product. The use of ratings or scales was suggested by several participants, with ‘Health Stars’ or ‘strength ratings’ similar to those used to indicate the intensity of coffee products mentioned. Scales combined with colors were suggested as a further improvement opportunity, with one participant specifically mentioning a ‘traffic light’ system.

Endorsement logos, such as the South African Heart Foundation logo, were highlighted as ‘easily seen’ and generally useful to make quick assessments, that is, facilitate a purchase decision without taking too long or requiring detailed nutritional knowledge. A national health endorsement logo with similar characteristics was also suggested. In contrast to a single positive health indicator, some participants suggested various types of health warning. Written warnings, such as those available on cigarette packaging, or highlighting ‘problematic’ ingredients such as sugar, were recommended. In addition, the use of bold text or information ‘written in red’ has been proposed to gain maximum consumer attention. The participants highlighted the need to base any warnings on holistic product characteristics using nutrient profile models (i.e., considering all nutrients and their ratios) in combination with the intended frequency of use.

The inclusion of health claims on products was also thought to be a useful intervention, informing consumers of the potential product-based health benefits, although possibly only benefitting ‘an already well-educated consumer’. Concern was expressed about the substantiation of the claims as well as ensuring strict criteria for inclusion on ‘appropriate products’ (i.e., health claims should not be allowed on energy-dense, nutrient-poor foods). The use of images of ‘teaspoons’ was a final suggestion that participants thought would serve as a simple yet effective indication of sugar content, but this was not mentioned in relation to other nutrients. Participants expressed concern about the lack of consistency in FOP labeling in the absence of a nationally endorsed format:


*“Whatever you do, you have to be consistent because consumers can’t see ‘live well, eat well’ for one company’s product [a local manufacturer’s logo], but then there are GDAs on all the multinationals’ products.”*
[PR21]

#### 3.2.3. Make it Smarter (Theme 3)

The final theme of label improvement revolved around the incorporation and use of technology. An illustrative quote from a consumer participant encapsulates this opportunity:


*“It is an enormous task to try and put a statement [on a package], a blanket statement that would fit 53 million people. But then I thought to myself—with algorithms and big data, it’s maybe not so far off anymore… Information interpreted for me, to help me to make better decisions.”*
[CN1]

Whilst suggestions to ‘make it simpler’ indicated several ways in which labels can be made easier to interpret for consumers, participants rightly highlighted the fact that it would be near impossible for a static, printed label to accurately meet the needs of a large, diverse population. As such, many participants spontaneously mentioned the potentially inclusive benefits of greater technological use on labels. A reference was made to innovative ways in which the South African government had ensured that information reached citizens (including those in rural areas) during the COVID-19 pandemic (i.e., through the mobile messaging application WhatsApp), and indicated that similar interventions could be considered for the dissemination of food and health-related information.

As emphasized by the participants, technology use is on the rise globally and in South Africa, with an increasing number of people owning smartphones. Through the use of mobile phones, several labeling interventions could be considered where the scanning of barcodes, QR codes, or other mechanisms could facilitate enhanced information transfer and use from labels. Many participants felt that technology-based interventions have the power to interpret food labels for consumers according to their differing needs or health conditions, thereby offering personalized recommendations. Furthermore, through gamification, technology-based food label initiatives can simultaneously enhance engagement with information or present it in more interesting ways.

However, not all participants agreed that technology poses a potential solution for improving the comprehension of food labels in South Africa. Valid concerns were that certain segments of the population may not own smartphones, and if they do, data would be expensive in South Africa, limiting affordability for a large section of the population. Furthermore, a concern was raised that technological interventions could leave less technologically savvy consumers behind—particularly those belonging to the ‘older generation’:


*“If your target is the poorly educated poor people, then technology is not the solution, because they can’t afford data… and the older generation, who didn’t grow up with that technology, get excluded.”*
[PR23]

#### 3.2.4. Challenges to Effective Static (On-Pack) Food Labels

Despite several suggestions for improvement, two key challenges relevant to all on-pack food labeling ‘solutions’ were raised by participants: (i) labels will not replace education about nutrition, and (ii) neither are we able yet to completely personalize them (even with the use of technology). There is still much unknown about the differing nutritional needs of consumers (illustrative quotes are included in Appendix A).

Labels are tools which consumers may use to facilitate decisions about food choices. Undoubtedly, the simplification of labels through the use of images or other means could make it easier for consumers to quickly assess the nutritional value of food. However, some participants expressed concern that over-simplification of labels may not be beneficial in the longer term, since it would not address the ultimate challenge of a lack of nutritional education. Participants generally agreed that nutrition education needs to be improved in South Africa and that more should be done to address this problem.

Whilst the use of technology highlighted under ‘make it smarter’ undeniably holds promise for the future, our current limitations regarding the availability of individualized nutrition data cannot be ignored. Participants were cognizant of this, as illustrated by references to the unclear role of the microbiome in good health and nutrition as well as different requirements regarding calorie intake. While we currently believe we have narrowed down ‘nutrients of concern’ for good health and wellbeing, we cannot truly be confident that the recommendations are valid for everyone. Furthermore, irrespective of how personalized technology may become in the future, its use could still be circumnavigated by consumers.

### 3.3. Quantitative Findings from the Consumer Survey

#### 3.3.1. Selection and Creation of FOP Labels

Based on the challenges mentioned or specific ideas for improvement suggested during the qualitative research phase, possible improvement initiatives were captured. Challenges and/or suggestions were cross-referenced with existing FOP labels. Based on the findings for Theme 1 (‘make it clearer’) and recommendations in Theme 2 (‘make it smarter’), the FOP labels of the fictitious product were made as follows: (i) large in size and (ii) printed in black and white. These criteria resulted in suggestions for multi-colored labels (e.g., traffic light labels) being eliminated from the potential pool of FOP labels that were tested with consumers, underpinned by the less time needed to detect when included on food labels compared with color versions [55].

While the suggestion to use teaspoons is a ‘concrete measure’, and a visual aid on pack is useful for sugar, it would not easily translate across other nutrients such as fat or protein, and was not considered for further development at this time. The resultant label categories selected for testing included warnings, health claims, GDA, and Health Star. During the interviews, participants expressed concern about their ability to discern between good and poor nutrient levels. As a result, two GDAs and two Health Star FOP labels were tested: ‘healthier’ and ‘less healthy’.

The health claims and warning FOP labels were considered antagonistic. For the health claim, it was decided to combine the concept of a single endorsement logo (for quick reference) with a more detailed health claim. A low GI claim was deemed appropriate considering that a fictitious cereal product was chosen for this study. The wording for the low GI claim was taken from South African draft legislation (R429) [56].

#### 3.3.2. Reliability of Survey Results

Both the HCS and the health rating data sets were considered reliable due to their high Cronbach alphas of 0.93 and 0.97, respectively (Appendix A).

#### 3.3.3. Performance of FOP Labels

Compared with the control (Product A), significant differences (*p* < 0.01) in the health rating of the cereal product were detected when different FOP labels were applied (Figure 4). For the remainder of the paper, the health ratings of the FOP labels, as per Figure 4, are referenced as the ‘main effect’.

Participants rated the cereal product with the warning FOP label (Product G) as the least healthy compared with the control (*p* < 0.01), with a very large effect size. Products with a low health star rating (Product E) were also rated as significantly less healthy (*p* < 0.01) than the control with a medium effect size. The GDA with a ‘less healthy’ nutritional profile (Product C) was rated as significantly less healthy (*p* = 0.03) compared to the control, but with a small effect size. Products bearing the endorsement logo/low GI claim (Product D) and high health star rating (Product B) were rated as healthier than the control (*p* < 0.01), but with a small effect size. The GDA with a ‘healthy’ nutritional profile (Product F) was not considered healthier than the control (*p* = 0.11). Except for Product F (‘healthier’ GDA profile), the FOP labels used in this study did guide the consumer to evaluate products as more/less healthy, although not with equal effect sizes.

Gender (interaction *p* = 0.03), income (interaction *p* < 0.01), and education (interaction *p* = 0.02) were found to affect the mean health ratings of Products A to G. However, in all cases, the trend followed a similar pattern as the main effect. Graphs illustrating the interaction are available in the Appendix A for gender, income, and education, respectively).

The points of difference for gender were: (i) women rated Product B as significantly healthier than the control, whereas men did not; and (ii) the mean health rating for Product G was lower for women than for men. In terms of mean product ratings (highest to lowest), the same pattern was obtained for male and female consumers, namely Products D, B, F, A, C, E, and G.

Overall, the estimated mean health ratings of the low-income group were higher than those of the high-income group (i.e., low-income earners gave higher ratings). However, both the low-income and high-income groups followed the same rating pattern as the main effect (highest mean rating to lowest): Products D, B, F, A, C, E, and G. The same trend was again observed for education levels; that is, respondents with lower education levels gave higher product ratings overall, but the ratings (highest to lowest) followed those of the main effect.

Race (*p* = 0.07), age (*p* = 0.68), smoking status (*p* = 0.42), physical activity level (*p* = 0.35), health consciousness (measured using the HCS; *p* = 0.11), and having children (*p* = 0.46) did not influence the health ratings of the product.

### 3.4. Integration

The joint display (Figure 5) integrates qualitative insights and survey results to propose a three-phase labeling strategy for South Africa. The phases are indicative of the timespan over which the changes can be implemented. For this study, the idea of concurrently managing both current and future opportunities for labeling resulted in three time horizons, with Horizon 1 being short-term, Horizon 2 being medium-term, and Horizon 3 being longer-term. The three qualitative themes (‘make it clearer’; ‘make it simpler’; and ‘make it smarter’) are thus positioned in the short, medium and long term. The identification of other inputs to the diagram is highlighted in the graph key as follows:Synopsis of the label improvement suggestions made by participants in the qualitative phase of the study;A single illustrative quote for the theme (additional quotes are available in the Appendix A);Application of qualitative insights to the consumer survey;A synopsis of the survey findings (exclusive to Horizon 2); andPositive and negative implications for each horizon through the integration of qualitative and quantitative data.

Through integration of the data, the positives (benefits) and negatives (outstanding challenges) to be considered for each horizon are highlighted (indicated by the corresponding positive or negative (5) for each horizon; Figure 5). No intervention was performed without any challenges. Whilst ‘make it clearer’ can be immediately implemented, the overall impact on consumers is likely to be small, considering that education on nutrition is still generally lacking. With the exception of the GDA on Product F, FOP labels guided consumers to evaluate products as more or less healthy relative to an unlabeled control (*p* < 0.01) (Horizon 2: (4); Figure 5). However, as the FOP labels did not address concerns about personalizing labels and technological interventions could not be tested with consumers in this study, these interventions are proposed for future research. Technological interventions may hold the key to decoding labels for consumers and personalizing information.

## 4. Discussion

Through the exploratory mixed-methods approach, this study provided deeper insights into the challenges that South Africans face when using food labels by investigating the usefulness of various FOP labels to help consumers judge the overall healthiness of food. Helping consumers to identify healthier foods is especially relevant in the South African context, which is highly culturally and linguistically diverse [18], and where the majority of the public is reliant on an under-resourced and fragmented healthcare system [3,57]. As an outcome of the research, short-term (Horizon 1), medium-term (Horizon 2), and longer-term (Horizon 3) strategic approaches, inspired by the time horizons of McKinsey [58], are suggested to concurrently manage both current and future opportunities for food labeling, potentially facilitating greater stakeholder alignment in the future. In the long term, improved food choices (made as a result of better understanding the nutritional content of packaged food, including through clearer and simpler food labels) could contribute to the reduction of the NCD burden in South Africa and thus add value to consumers and the government alike. The three overarching phases and the corresponding opportunities based on the time horizons are discussed in this section.

### 4.1. Short Term: Make It Clearer

Consistently, small and illegible fonts on food packaging remain a challenge in South Africa [10,13,14,15,16,17], despite no legal limitations on enlarging the text or enhancing legibility [59]. Food manufacturers can perform better by choosing to print black-on-white (or vice versa) and increasing the font according to pack size. This intervention will not apply to small packages (total exterior area of 2000 mm^2^ or less) [59], but this is expected to be a minority of products (e.g., chewing gum). For progressive manufacturers, no legal limitation prevents immediate implementation of such an initiative; however, without legislative guidelines, it is unclear what percentage of the market would feel compelled to ‘make it clearer’. The food industry has a poor reputation for taking positive health initiatives through self-regulation [60,61,62] and has been implicated in influencing policy processes in many countries [63,64,65,66]. Unfortunately, this intervention will still not aid consumers who lack basic nutritional knowledge to interpret information on food labels. In a country such as South Africa, where health literacy and access to healthcare are vastly unequal [3,67,68], the expected benefits from such a change may still be few.

Despite the apparent limitations of ‘make it clearer’, these recommendations (to increase font size and improve legibility) were applied in the design of the FOP labels of the fictitious cereal product. The FOP labels were designed to cover approximately one-eighth of the total front-facing of the fictitious cereal pack. This is similar to the label area dedicated to health warnings on the back label of liquor products in South Africa [69], but smaller than that of tobacco products, where warnings cover 15% of the front and 25% of the back of the pack [70]. If South Africa moves to the use of FOP labels in the future, a minimum size limit should be considered to prevent them from being too small to be legible.

The use of black-and-white contrast should be considered to ensure good contrast and further enhance the legibility. Owing to variations in packaging materials (e.g., metallized foil compared to white liner carton), legibility cannot otherwise be guaranteed. Black-and-white labels not only benefit legibility but are also cheaper to print. The use of black-and-white FOP labels will presumably not necessitate additional costs for food manufacturers, as no additional colors (e.g., green, red, orange/yellow) would be required. Additionally, the use of black-and-white FOP labels (as opposed to color) may present a lower point of resistance from the manufacturers.

### 4.2. Medium Term: Make It Simpler

From the qualitative interviews, it was clear that professionals and consumers alike believe that South African food labeling can be simplified to improve understanding. The range of suggestions generally aligned with FOP labeling used in other countries: GDAs, first used in the United Kingdom [71], indicate nutrients and relative percentages; health stars used in Australia and New Zealand [24] aid consumers in distinguishing foods on a scale from 1 to 5; warnings are mandated in Chile [39]; and health claims may be used upon successful application in the European Union [72].

Research has indicated that the usefulness of GDAs is generally limited [21,73]. While Product C was considered significantly healthier than the control, the effect size was small, and Product F could not be distinguished from the control (Product A). Although there are no mandates requiring GDAs in South Africa, these FOP labels are present on some multinational products, along with variations created by manufacturers or retailers. This coexistence of many FOP labeling systems in the marketplace can be confusing to consumers [74,75,76], and highlights the importance of arriving at a single, agreed FOP label. Based on the findings of this study, the further and future use of GDAs is not supported as a solution for South Africa.

Consumers were able to effectively make appropriate health assessments for products bearing a health claim (Product D), warning (Product G), and Health Stars (Products B and E). Product G had the largest effect size, which was attributed to the warning label, followed by the low Health Star rating (Product E: medium effect size). However, these systems are not directly comparable, because they have different aims [77]. Briefly, a key difference is that the Health Star Rating system awards points for positive food components, whereas the warning system does not. Based on the results of this study, only health stars present a means by which consumers can effectively identify both less healthy foods and those that may be considered healthier. Similar recommendations exist to include FOP labels for both healthy and unhealthy foods [78]. Both health stars and warnings lack reference points [79]. It is uncertain what consumers would think of foods that carry no health warning; that is, would such food automatically be considered healthy, and if so, how healthy? Unintended consequences of health warnings on food also merit further research [80], especially in developing countries such as South Africa, where hunger and food insecurity are rife [81,82].

Further real-life studies such as those conducted by Dubois et al. [83] on both warning and Health Star labels are recommended. In the present study, both types of FOP labels accurately guided consumers to identify products as more or less healthy, but as highlighted by interview respondents, they do not replace nutrition education. Furthermore, FOP labels can be ignored. Ikonen et al. [84], applying interdisciplinary meta-analysis, found that although FOP labels aid consumers in identifying healthier products, their capacity to nudge consumers towards healthier purchasing choices is more limited. An et al. [85] came to similar insights through a systematic review, specifically highlighting that while FOP labels may assist in the identification of healthier products, there is no clear evidence for subsequent altered food purchases. FOP labels may lead to halo effects, positively influencing ‘healthier’ (virtue) as well as ‘less healthy’ (vice) products, but they only influence the purchase intention of virtue products [29,84].

### 4.3. Long Term: Make It Smarter

A challenge with FOP labels is that they are not personalized to individual needs. A one-size-fits-all approach to nutrition is not appropriate for everyone, and more personalized interventions are likely to be more effective in achieving the desired health outcomes, including a reduced incidence of NCDs [86,87]. Ultimately, dietary information should be better tailored to personal needs to enable informed food choices [88,89].

New technologies, encompassing a multitude of features and enabling greater levels of personalization, are emerging at increasing frequency [90,91], leaving no doubt that the food labels of the future will be smarter. The Internet of Things (IoT) is becoming increasingly noticeable in our daily lives [92] and smartphone penetration is rising rapidly, even in South Africa, where five million additional users are expected in the 12 months to 2023 [93]. Furthermore, older adults can be successfully encouraged to adopt technology when value or personal relevance is evident [94,95,96].

Predictions for the next decade include ‘universal internet’, where the challenges of bad connections and internet costs may become redundant [97]. Directly incorporating technological ‘solutions’ into food labels should be considered. More personalized information could be made available for consumers to use, as and when needed. While there is currently a lack of personalized data about consumer needs and preferences [98], in future, this could be overcome by solutions such as crowdsourcing [99]. Engaging the adolescent population in obtaining innovative ideas to promote healthy eating is an approach used in Latin America [100]. The importance of social media and the involvement of celebrities and influencers present more avenues for change.

It has been proposed that developing countries do not have any alternatives to technology adoption other than leapfrogging new and advanced technologies [101]. Food labeling presents an area where it could be advantageous for South Africa to opt for a technology-centric approach, since this has proven successful in other areas of Africa, albeit in other disciplines. One such example is M-Pesa, a mobile phone-based money-transfer service in Kenya. Through the unique approach of ‘trading with airtime’ and using an ‘agent network’, over 86% of adult Kenyans are today financially included, compared to 26.7% when M-Pesa was launched 12 years ago. Thus, it is not unthinkable to enable nutrition education inclusion through technology in a country such as South Africa.

Indeed, access to technology is currently a hurdle for some South Africans; however, this can be expected to change in the future. Designing for the future and making labels smarter should be an urgent avenue of focus for South African researchers and policymakers. Using technology, it may be possible to respond to consumers’ varying expectations and information needs [102,103] in ways that have never been possible before. To inform purchase choices, labeling (whether digital or otherwise) must be both useful and easy to use [104]. Personalized, preventive dietary guidance (enabled through technology) that is seamlessly integrated with consumer routines that take socio-economic challenges and preferences into account, as well as challenges related to language inclusivity [17], could be critical to solving the public health challenges we face in South Africa.

### 4.4. Limitations

A limitation of this study is the fact that professional contacts of the lead author were approached for the interviews (accounting for approximately 50% of professional participants). However, the authors believe that the results incorporate highly varied feedback without a vested interest in one FOP label format dominating. Furthermore, choices made throughout the study were rationalized. We feel that our transparency on this matter is illustrative of good research ethics.

The study did not evaluate all possible FOP label formats, and so this is another limitation of the present study. Additional formats, such as the Nutri-Score and other colored labels, should be considered in future studies. All FOP labels are currently limited one way or another and, unfortunately, nothing will replace nutritional education. We are, however, hopeful that with advances in personal nutrition and technology a less confusing food and nutrition future awaits us.

Finally, conducting the online consumer survey could have excluded respondents who were not Internet enabled. One product type (cereal) was selected for this survey, so the results may only be valid for packaged cereal products. Therefore, consumer perceptions of other product categories must be verified on a case-by-case basis. Furthermore, these findings do not predict how people would react in real-life purchasing situations.

## 5. Conclusions

The aim of this research was to establish what labeling interventions can help South African consumers to better comprehend the healthiness of foods. This included investigating the extent to which various FOP labels can help consumers to judge the overall healthiness of a fictitious cereal product, a commonly consumed ultra-processed food type in South Africa. Both a health warning and Health Star-like FOP label hold promise to help consumers gauge the overall healthiness of food products. Unfortunately, neither of these FOP labels can be personalized to individual requirements at present, nor will they replace nutrition education. Through better understanding of the healthiness of food, consumers will be empowered to make healthier food choices, which can help to reduce the NCD burden. Real-life studies using physical products and behavioral measures of food selection (i.e., choice) and consumption are recommended.

While the use of FOP labels in South Africa could be a viable policy option to guide consumers to reduce their consumption of ‘less healthy’ foods and encourage the consumption of ‘healthier’ products, it should form part of a broader policy strategy. As part of that policy strategy, researchers and policymakers alike must prioritize the investigation of more forward-thinking, technology-centric ‘solutions’.

## Figures and Tables

**Figure 1 nutrients-14-02801-f001:**
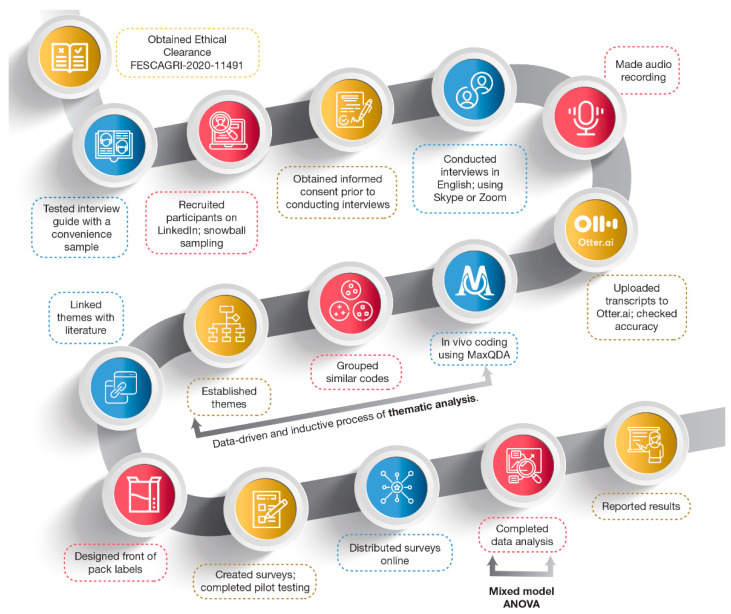
Exploratory mixed-methods sequential study process (ANOVA: Analysis of variance).

**Figure 2 nutrients-14-02801-f002:**
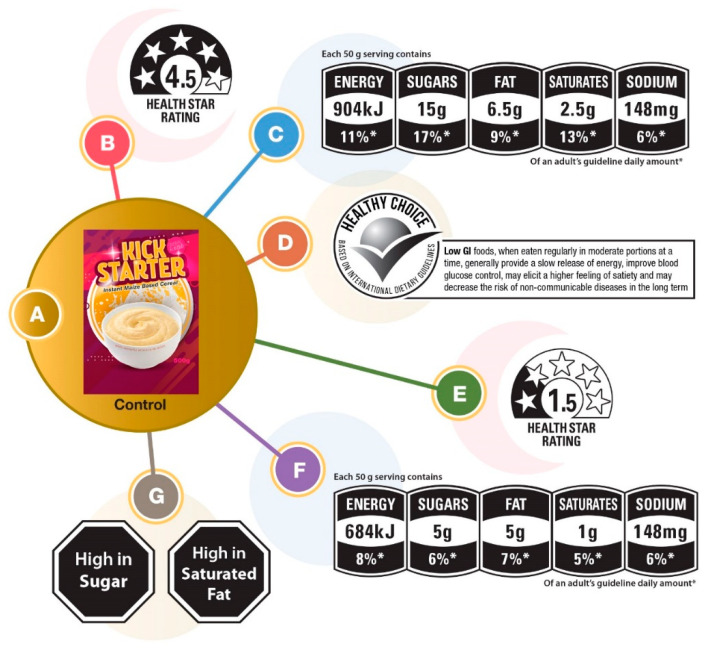
Packaging design of fictitious cereal product and FOP labeling used in the consumer survey. *: of an adult’s Guideline Daily Amount.

**Figure 3 nutrients-14-02801-f003:**
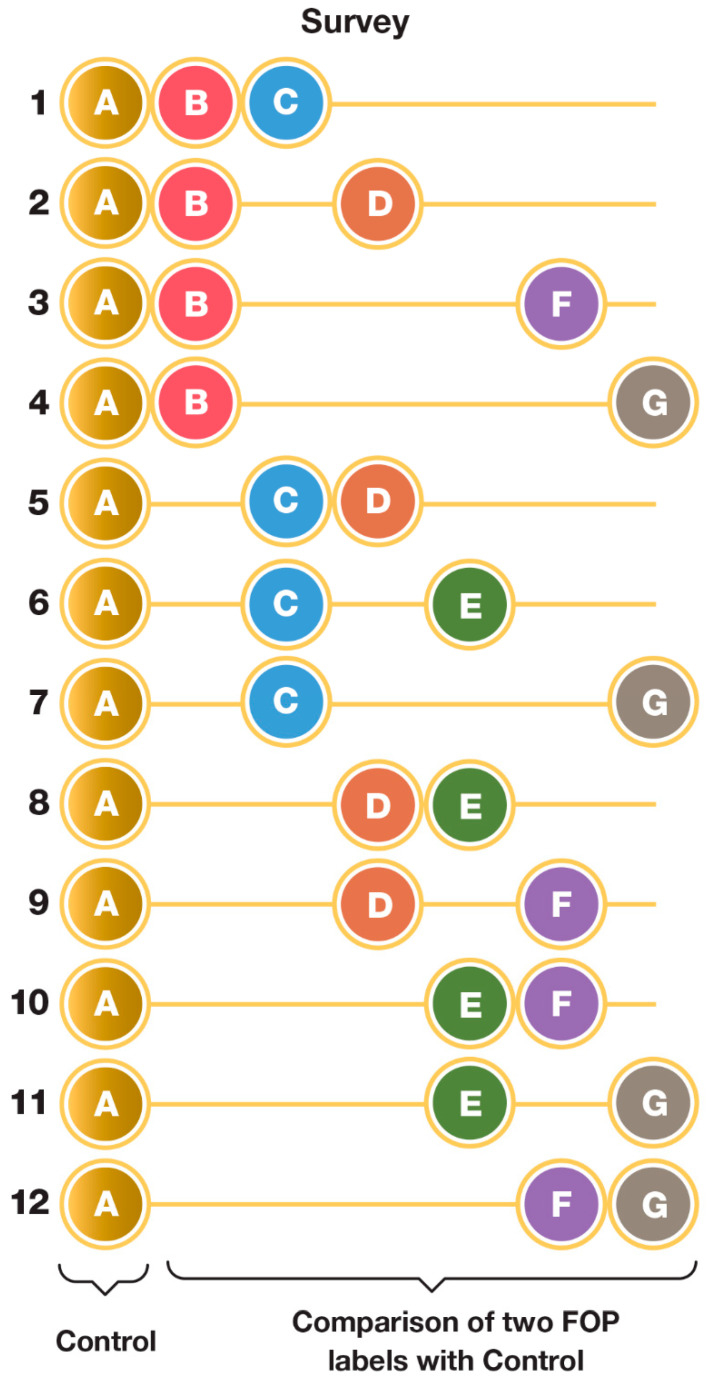
Design of the twelve surveys, with each including the ‘no-label’ control and a different combination of two FOP labels. Graphical details of symbols A to G as depicted in Figure 2.

**Figure 4 nutrients-14-02801-f004:**
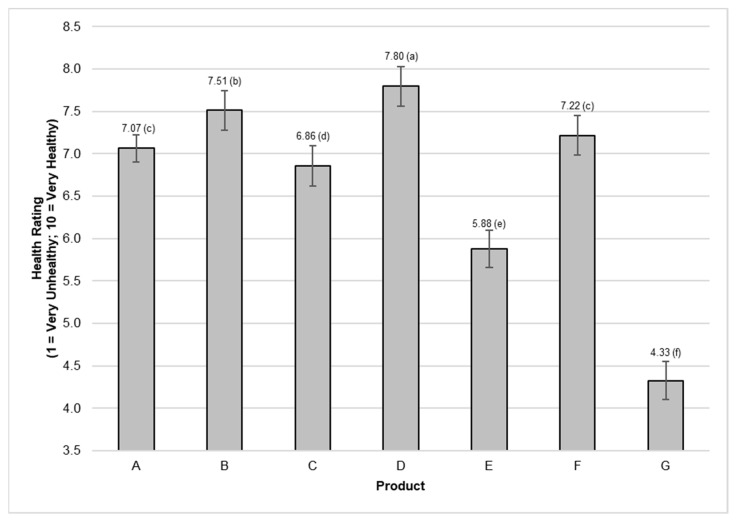
Label ratings for fictitious cereal product with different FOP labels. Product A: control; Product B: high Health Star Rating; Product C: Guideline Daily Amount with a ‘less healthy’ nutritional profile; Product D: endorsement logo/low Glycemic Index claim; Product E: low Health Star Rating; Product F: Guideline Daily Amount with a ‘healthy’ nutritional profile; Product G: warning. Differences with a significance level of 5% (*p* < 0.05) were considered statistically significant and are indicated by different alphabetical letters on the graph (a–f).

**Figure 5 nutrients-14-02801-f005:**
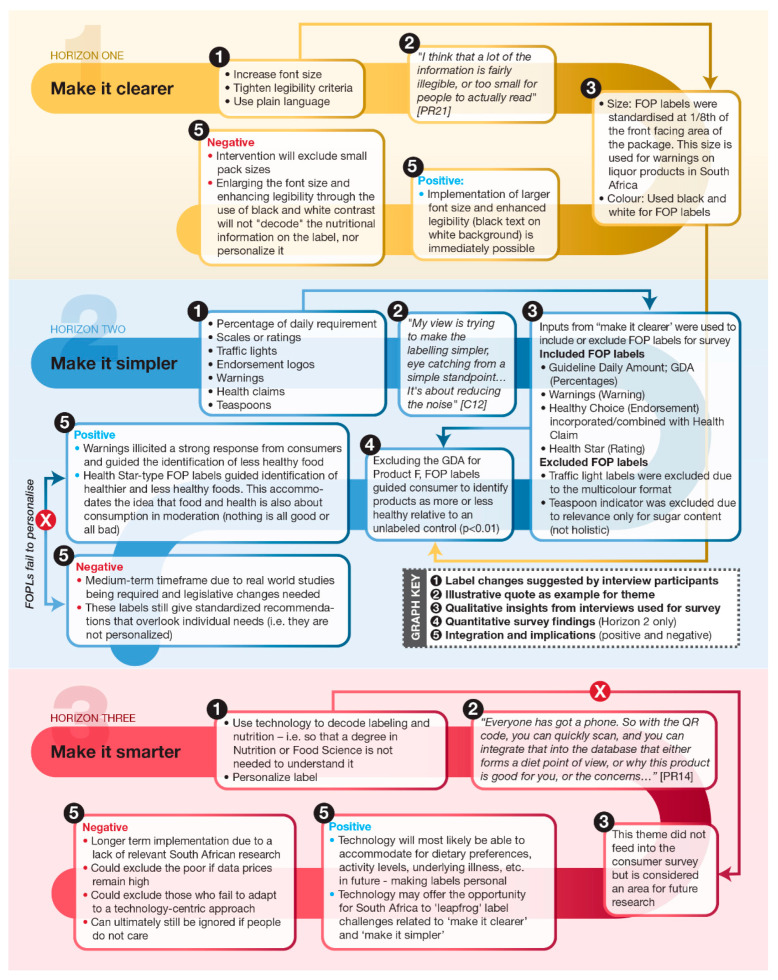
Joint display integrating interview themes with survey outcomes to illustrate the positives and negatives of a staged approach to implementing FOP labeling in South Africa. FOP: Front-of pack.

**Table 1 nutrients-14-02801-t001:** Characterization of the consumer survey sample: Sociodemographic characteristics, self-reported health circumstances and food-shopping behavior (*n* = 1261).

Demographic Attribute	Category	Total (*n*)	ercentage (%)
**Gender**	Female	915	72.6
	Male	346	27.4
**Age**	18–24 years old	149	11.8
	25–34 years old	493	39.1
	35–49 years old	441	35.0
	50+ years old	178	14.1
**Race**	Black	730	57.9
	Colored	156	12.4
	White	274	21.7
	Indian/Asian	93	7.4
	Prefer not to answer	6	0.5
	Other	2	0.2
**Household Income (monthly)**	Less than ZAR 5000	365	28.9
	ZAR 5000–ZAR 9999	239	19.0
	ZAR 10,000–ZAR 19,999	232	18.4
	ZAR 20,000–ZAR 29,999	155	12.3
	ZAR 30,000–ZAR 69,999	203	16.1
	ZAR 70,000+	67	5.3
**Education**	Less than Grade 12	94	7.5
	Grade 12	394	31.2
	Trade or vocational training	247	19.6
	Diploma	253	20.1
	Degree or Postgraduate Degree	273	21.6
**Children (<18 years) in Household**	0	356	28.2
	1	340	27.0
	2	359	28.5
	3	147	11.7
	4+	59	4.7
**Household size (Total)**	1	74	5.9
	2	206	16.3
	3	248	19.7
	4	286	22.7
	5	213	16.9
	6	116	9.2
	7+	118	9.4
**Province of residence**	Gauteng	615	48.8
	Western Cape	225	17.8
	KwaZulu-Natal	188	14.9
	Rest of South Africa	233	18.5
**Industry of employment**	Arts, Entertainment, Recreation	44	3.5
	Unemployed	263	20.9
	Education	102	8.1
	Food	58	4.6
	Financial Services	73	5.8
	Government, Public Administration	92	7.3
	Healthcare	65	5.2
	Media, Advertising, Public Relations	68	5.4
	Mining, Construction	60	4.8
	Scientific or Technical Services	22	1.7
	Student	94	7.5
	Telecommunications	42	3.3
	Other	278	22.0
**Relationship status**	Single	440	34.9
	In a relationship	359	28.5
	Married	387	30.7
	Divorced	56	4.4
	Other	19	1.5
**Smoking status**	Non-smoker	800	63.4
	Smoker	229	18.2
	Occasional smoker	122	9.7
	Ex-smoker	110	8.7
**Physical activity level**	Inactive	297	23.6
	Less than 150–300 min moderate-intensity exercise OR 75–150 min high-intensity exercise in a week	511	40.5
	About 150–300 min moderate-intensity exercise OR 75–150 min high-intensity exercise in a week	327	25.9
	More than 150–300 min moderate-intensity exercise OR 75–150 min high-intensity exercise in a week	126	10.0
**Shopping responsibility**	Shared responsibility	430	34.1
	Sole responsibility	831	65.9
**Cereal purchased**	Occasional/Sometimes	171	13.6
	Yes	1090	86.4
**How well do you understand food labels?**	Not at all	45	3.6
	A little	110	8.7
	Fifty-fifty	268	21.3
	Fairly well	391	31.0
	Very well	447	35.4
**Do food labels inform your food choices?**	Not at all	91	7.2
	A little	126	10.0
	Fifty-fifty	297	23.6
	Fairly well	362	28.7
	Very well	385	30.5
**Most searched nutrients on food labels (all that apply)**	Sugar	897	71.1
	Energy	849	67.3
	Fat	787	62.4
	Carbohydrates	752	59.6
	Protein	734	58.2
	Sodium	510	40.4
	Do not use label at all	113	9.0
**Main factor influencing food choices**	Health	521	41.3
	Price	367	29.1
	Taste	218	17.3
	Brand	126	10.0
	Appearance	23	1.8
	Other	6	0.5
**Main source of health information**	Food labels	389	30.8
	Doctor or dietician	263	20.9
	Social Media	214	17.0
	Books and Magazines	165	13.1
	Friends and Family	129	10.2
	Television or radio	57	4.5
	Other	44	3.5

## Data Availability

The interview and survey data (including interview guide, survey questions, anonymized transcripts, survey images and survey response data) presented in this study are available upon request to the corresponding author.

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
