# Peer review of "Healthy or Not Healthy? A Mixed-Methods Approach to Evaluate Front-of-Pack Nutrition Labels as a Tool to Guide Consumers"

_nutrients, 2022, doi:10.3390/nu14142801_

Round 1
Reviewer 1 Report
Thank you very much for the opportunity to read this manuscript.
FOP nutrition labels are essential to health programs encouraging consumers to make informed and “good choices”. The key is that the labels should convey an unambiguous message. So that only one glance allows consumers to make the right decisions. Moreover, I agree with the Authors that it is even more critical in a country with so much linguistic and cultural diversity.
This study has a correct methodology (although I must admit that it is a bit confusing - could be broken down into two publications.) Also, the manuscript is very well written in English.
My biggest objection and disappointment is the exclusion of colored labels. First, the Authors’ statements (lines 501-508) are inaccurate - originally colored labels can also be printed in grayscale without losing information potential. Secondly, the mentioned traffic light system is becoming more common and will soon become mandatory throughout the European Union. Therefore, it should be assumed that many products on the South African market will be labeled with, for example, Nutri-Score. Thus, excluding it from this analysis undermines its sense and usefulness.
Moreover, this (line 118-123) way of recruiting raises my far-reaching concerns about objectivity.
Do I understand correctly that there was only one physician in this group?
Unfortunately, these “errors” cannot be resolved, so if the Editors want to publish this manuscript, I suggest that the Authors clearly emphasize these limitations in the discussion.
Reviewer 2 Report
The article is interesting but I don't know if it is in fact included in the scope of this journal. I understand that the context is necessary to evaluate the characteristic of a food, mainly due to the increase in non-communicable chronic diseases, but I could not find this link with these diseases.
There is more of an interest focused on labels than an association with diseases. I believe that this should be improved, since the authors make this association in the article, especially in the keywords and in a part of the introduction (lines 32-40).
Round 2
Reviewer 1 Report
I must admit that I very positively received the Authors' answers, the way of argumentation, and the changes introduced in the manuscript (especially in terms of emphasizing the study's limitations). I can see that the Authors are honest and straightforward, and the manuscript has gained a lot despite my initial concerns. Thank you very much.Reviewer 2 Report
The manuscript must be accepted.